Sexual and psychological health of couples with azoospermia in the context of the COVID-19 pandemic

Dong Meng 1 2 3
Tao Yanqiang 4
Wu Shanshan 1 2
Li Zhengtao 1
Wang Xiaobin 1
Tan Jichun tanjc@sj-hospital.org 1 2
1 Center of Reproductive Medicine, Shengjing Hospital of China Medical University , Shengyang , Liaoning , China
2 Key Laboratory of Reproductive Dysfunction Diseases and Fertility Remodeling of Liaoning Province , Shenyang , China
3 School of Life Sciences, China Medical University , Shenyang , Liaoning , China
4 Faculty of Psychology, Beijing Key Laboratory of Applied Experimental Psychology, Beijing Normal University , Beijing , China
Vonk Jennifer
Electronic publication date: 2021 Oct 20
Publication date: 2021
Volume: 9
Electronic Location ID: e12162
Received 2021 May 11; Accepted 2021 Aug 25
Copyright: ©2021 Dong et al.
Copyright year: 2021
Copyright holder: Dong et al.
License: This is an open access article distributed under the terms of the Creative Commons Attribution License, which permits unrestricted use, distribution, reproduction and adaptation in any medium and for any purpose provided that it is properly attributed. For attribution, the original author(s), title, publication source (PeerJ) and either DOI or URL of the article must be cited.
License URL: https://creativecommons.org/licenses/by/4.0/

Keywords: COVID-19, Couples with azoospermia, Sexual health, Psychological health

Funding: The National Key Research and Development Program 2018YFC1002105 The Major Special Construction Plan for Discipline Construction Project of China Medical University 3110118033 The Shengjing Freelance Researcher Plan of Shengjing Hospital of China Medical University This study was supported by a grant from the National Key Research and Development Program (2018YFC1002105), the Major Special Construction Plan for Discipline Construction Project of China Medical University (3110118033), and the Shengjing Freelance Researcher Plan of Shengjing Hospital of China Medical University. The funders had no role in study design, data collection and analysis, decision to publish, or preparation of the manuscript.

==============================
Background

To date, there have been no reports on the sexual and psychological health of patients with azoospermia during the coronavirus disease 2019 (COVID-19) pandemic. Previous studies on the sexual health of couples with azoospermia are limited and are especially lacking in data on the wives of azoospermic men.

Methods

We conducted a case–control cross-sectional study between 1 July 2020 and 20 December 2020. In total, 100 couples with azoospermia comprised the experimental group and 100 couples with normozoospermia comprised the control group. The couples’ sexual health was measured using standardised sexual function questionnaires (male: International Index of Erectile Function-15 [IIEF-15] and Premature Ejaculation Diagnostic Tool [PEDT]; female: Female Sexual Function Index [FSFI]) and a self-designed questionnaire to evaluate changes in sexual behaviours (sexual satisfaction, desire, frequency of sexual activity, masturbation, and pornography use) during lockdown. The couples’ psychological health was measured using the 7-item Generalized Anxiety Disorder (GAD-7) scale and 9-item Patient Health Questionnaire (PHQ-9). The Actor–Partner Interdependence Model (APIM) was used to analyse the associations between sexual health and psychological health.

Results

The IIEF-15 scores (53.07 ± 11.11 vs. 57.52 ± 8.57, t =  − 3.17, p = 0.00) were lower and the PEDT scores (6.58 ± 3.13 vs. 5.17 ± 2.22, t = 3.67, p = 0.00) and incidence of premature ejaculation (χ2 = 14.73, p = 0.00) were higher for men with azoospermia than for men with normozoospermia. Compared with those of wives of men with normozoospermia, the total FSFI scores (25.12 ± 5.56 vs. 26.75 ± 4.82, t =  − 2.22, p = 0.03) of wives of men with azoospermia were lower. The chi-square test showed that the perceived changes in sexual satisfaction (χ2 = 7.22, p = 0.03), frequency of masturbation (χ2 = 21.96, p = 0.00), and pornography use (χ2 = 10.90, p = 0.01) were significantly different between the female groups with azoospermia and normozoospermia, but there were no significant changes in sexual behaviour between the male groups. The GAD-7 (men: 7.18 ± 5.56 vs. 5.68 ± 4.58, p = 0.04; women: 6.65 ± 5.06 vs. 5.10 ± 3.29, p = 0.01) and PHQ-9 scores (men: 10.21 ± 6.37 vs. 7.49 ± 6.10, p = 0.00; women: 8.81 ± 6.50 vs. 6.98 ± 4.43, p = 0.02) were significantly higher for couples with azoospermia than for couples with normozoospermia. The APIM showed that for couples with azoospermia, sexual function negatively correlated with their own anxiety (men: β = −0.22, p = 0.00; women: β = −0.38, p = 0.00) and depression symptoms (men: β = −0.21, p = 0.00; women: β = −0.57, p = 0.00) but not with their partner’s anxiety and depression symptoms (p > 0.05).

Conclusions

Couples with azoospermia had a lower quality of sexual function and higher levels of psychological distress than couples with normozoospermia. Their sexual health negatively correlated with psychological distress.

Introduction

The coronavirus disease 2019 (COVID-19) pandemic has brought about challenges to global healthcare and has had a significant effect on individual psychological health and quality of life (Brooks Samantha et al., 2020; Huang et al., 2020). The COVID-19 pandemic may have contributed to increased rates of depression, anxiety, and post-traumatic stress disorder in the general population and in COVID-19 survivors (Dutheil, Mondillon & Navel, 2021; González-Sanguino et al., 2020; Liu et al., 2020; Shuja et al., 2020). Sexual activity is an essential part of life (Davison et al., 2009), and sexual health is directly related to mental health and quality of life (World Health Organization, 2002; World Association for Sexual Health, 2006). Based on the psychological distress caused by this pandemic, it is expected that sexual health would also be affected (Ballester-Arnal et al., 2020). Recent investigations on the effects of the COVID-19 lockdown on sexual health have shown reduced quality of sexual life (Fuchs et al., 2020; Li et al., 2020a; Li et al., 2020b; Yuksel & Ozgor, 2020).

Male infertility is a complex multifactorial disease, affecting 7–12% of men in the general population (Forti & Krausz, 1998; Lotti & Maggi, 2018). Azoospermia is the most serious form of infertility, affecting approximately 1% of men (Tournaye, Krausz & Oates Robert , 2017). Azoospermia is defined as at least two microscopic analyses confirming the absence of spermatozoa after centrifugation of the complete semen sample (Practice Committee of the American Society for Reproductive Medicine in collaboration with the Society for Male Reproduction and Urology, 2018). Being diagnosed with azoospermia can be a devastating experience for infertile couples (Johansson, Hellström & Berg, 2011). Couples with azoospermia who are aware that sexual intercourse cannot lead to pregnancy (‘firing blanks’) may experience both depressive symptoms and somatic anxiety (Lotti et al., 2016). For men with azoospermia, sperm can be obtained only through surgical procedures, such as testicular sperm extraction or epididymal sperm extraction (Song et al., 2020). If the procedure fails, they will not be able to conceive a biological child. Furthermore, the fertilisation and clinical pregnancy rates for couples who attempt to conceive with sperm obtained through a surgical procedure are significantly lower than those for couples with normozoospermia (Vloeberghs et al., 2015).

The unfulfilled and strong desire to have children causes significant anxiety among couples with azoospermia, which may have worsened during the COVID-19 pandemic because of the interruption in fertility treatments owing to the lockdown. This interruption in fertility treatments, an uncontrollable and stressful event, has caused increased anxiety and depression in patients with infertility (Ferrero et al., 2020; Boivin et al., 2020; Ben-Kimhy et al., 2020). Stress related to the lockdown, postponed fertility care, and uncertainty can affect mental well-being, especially in men with azoospermia, whose pre-pandemic levels of distress and functional impairment are typically higher than those of patients with other types of infertility (Bechoua, Hamamah & Scalici, 2016).

There are few studies focusing on the sexual health of couples with azoospermia, but they lack data on the wives of men with azoospermia (Lotti et al., 2016). Therefore, an investigation of the sexual health of couples with azoospermia is warranted. This study primarily aimed to investigate the sexual health (including sexual function and perceived changes in sexual behaviours during lockdown) and psychological health (including anxiety and depression status) of couples with azoospermia compared with those of couples with normozoospermia. Moreover, we aimed to explore the association between sexual health and psychological health in couples with azoospermia using the Actor–Partner Interdependence Model (APIM).

Materials and Methods

Participant selection

A case–control cross-sectional questionnaire survey was administered at the Center of Reproductive Medicine of Shengjing Hospital of China Medical University between 1 July 2020 and 20 December 2020. The experimental group consisted of men with azoospermia and their wives, whereas the control group consisted of men with normozoospermia (according to WHO semen examination standards, (World Health Organization, 2010) and their wives. All participants were recruited from our Reproductive Medicine Center. The semen test results of the men in the control group was normal (men with normozoospermia), and no spouse was diagnosed with infertility. Men taking drugs (e.g., selective serotonin reuptake inhibitors, tricycle antidepressants, and phosphodiesterase type 5 inhibitors) that may affect their ejaculatory and erectile function and/or mental state were excluded (Gao et al., 2013). All participants were aware of their fertility status (including semen status) before completing the questionnaire. Before enrolment in the study, all participants were informed of the voluntary and anonymous nature of the study design and that their privacy would be protected. None of the participants had any biological children.

The questionnaires were completed in a private room at the Center of Reproductive Medicine of Shengjing Hospital of China Medical University to fully ensure participants’ privacy, and no spouses were present. Participants were not compensated in any form. Participants who provided inconsistent answers were excluded (e.g., in the questionnaire focused on sexual function, ‘did not attempt intercourse’ was selected in one question but not others).

Measures

We used a self-designed questionnaire and five standardised questionnaires, all of which were validated in the Chinese language.

Demographic variables

Data on the patients’ baseline information, including age, height, weight, occupation, educational level, work or life stress levels, and lifestyle habits (such as frequency of physical exercise, smoking status, and drinking status) were collected.

Men’s sexual function

Men’s sexual function was assessed using the International Index of Erectile Function-15 (IIEF-15) and Premature Ejaculation Diagnostic Tool (PEDT) and was based on their sexual status in the preceding 4 weeks. The IIEF-15 includes 15 items covering 5 domains of male sexual function (erectile function, orgasmic function, sexual desire, intercourse satisfaction, and overall satisfaction). The total score ranges from 6 to 45, with Cronbach’s alpha values ≥ 0.91 (Rosen et al., 1997; Corona, Jannini & Maggi, 2006). The presence and severity of ED were based on the IIEF-15 score, with the severity of ED represented as follows: 26–30, none; 22–25, mild; 17–21, mild to moderate; 11–16, moderate; and <11, severe (Cappelleri et al., 1999).

The PEDT includes five items, which are scored on a 5-point scale from 0 to 4, with a total severity score ranging from 0 to 20. A PEDT score ≤ 8 indicated no premature ejaculation (PE), 9–10 indicated probable PE, and ≥11 indicated implied PE. The Cronbach’s alpha value was 0.78 (Symonds et al., 2007).

Women’s sexual function

Female sexual function was assessed using the Female Sexual Function Index (FSFI), which includes 19 items and 6 domains of sexual function (desire, arousal, lubrication, orgasm, satisfaction, and coital pain) based on the sexual status in the preceding 4 weeks. The scores for each domain range from 1.2 to 6 or from 0 to 6, and the total score ranges from 2 to 36, with Cronbach’s alpha value ≥ 0.82 (Rosen et al., 2000). A total FSFI score ≤ 23.45 (Chinese cut-off) indicated that the woman might have sexual dysfunction (Ma et al., 2014; Lo & Kok, 2018).

Sexual behaviours during lockdown

We measured five aspects of self-reported changes in sexual behaviours during the lockdown using five questions, which were adopted from previous studies’ questionnaires (Li et al., 2020a; Li et al., 2020b). Change in sexual desire was assessed by asking the following question: ‘Compared with that before the COVID-19 pandemic, how has your sexual desire changed’? Change in sexual frequency was assessed by asking the following question: ‘Compared with that before the COVID-19 pandemic, how has your sexual frequency changed’? Change in sexual satisfaction was evaluated by asking the following question: ‘Compared with that before the COVID-19 pandemic, how has your sexual satisfaction changed’? The answers to the above three questions were set as follows: increased, unchanged, and decreased. Change in frequency of masturbation was assessed by asking the following question: ‘How has your frequency of masturbation changed compared with that before the COVID-19 pandemic’? Change in frequency of pornography use was evaluated by asking the following question: ‘How has your frequency of pornography use changed compared with that before the COVID-19 pandemic’? The answers to the above two questions were set as follows: none, increased, unchanged, and decreased.

Psychological health

Anxiety symptoms were evaluated using the 7-item Generalized Anxiety Disorder scale (GAD-7). The items are scored on a 4-point scale, from 0 to 3, and the total score ranges from 0 to 21. The cut-off score of ≥ 10 was used to assess the presence of anxiety (Löwe et al., 2008).

Depression symptoms were evaluated using the 9-item Patient Health Questionnaire (PHQ-9), with Cronbach’s α value ranging from 0.73 to 0.95. The items are scored on a 4-point scale from 0 to 3, with a total severity score ranging from 0 to 27. The cut-off score of ≥ 10 was used to assess the presence of depression (Kroenke, Spitzer & Williams, 2001).

Additionally, a question on COVID-19-related anxiety (‘Does the novel coronavirus pneumonia pandemic make you anxious’) was also asked, in which respondents rated their anxiety as severe, slight, or none.

Statistical analyses

Data analyses were performed using the statistical software Statistical Package for the Social Sciences (version 22.0; International Business Machines Corp., Armonk, NY, USA). Results are presented in a tabular format. Categorical variables are summarised as counts and percentages, in addition to continuous measures with counts, means, and standard deviations (SDs). The chi-square test was used to compare categorical data, and the independent t-test and one-way analysis of variance (ANOVA) were used to compare numerical data. Effect sizes were used to evaluate the strength of each statistical analysis and were measured with Cohen’s d for independent t-tests, and Cramer’s V for chi-square (χ2) tests.

We used the APIM (Cook William & Snyder Douglas, 2005) to assess the association between sexual function and psychological health for both sexes. We used the R package ‘lavaan’ for calculating the mediation effects. The actor effect was defined as the effect of an individual’s sexual function on his/her own depression and anxiety symptoms, and the partner effect was defined as the effect of an individual’s sexual function on his/her partner’s depression and anxiety symptoms (Fernandes et al., 2021). The overall fitting model and goodness-of-fit were evaluated with the following indices: degrees of freedom values (χ2/df) < 3, root mean square error of approximation (RMSEA) < 0.10, standardised root mean square residual (SRMR) < 0.10, normed fit index (NFI) > 0.90, and comparative fit index (CFI) > 0.90 (Ryu, 2014).

A two-tailed p value <0.05 indicated statistical significance for all tests.

Ethical approval

The study protocol was approved by the Institutional Review Board for Research on Human Subjects at the Center of Reproductive Medicine (approval number 2020PS009F).

Results

Participant responses and demographic characteristics

A total of 128 couples (men diagnosed with azoospermia and their wives) belonging to the experimental group participated in the study; 28 couples were excluded because they provided inconsistent answers and/or failed to answer all questions (18 failed to answer all questions and 10 provided inconsistent answers). A total of 135 couples (men diagnosed with normozoospermia and their wives) belonging to the control group participated in the study; 35 couples were excluded (23 failed to answer all questions and 12 provided inconsistent answers). Ultimately, the responses of 200 couples (100 couples with azoospermia, 100 couples with normozoospermia) were included in the analysis (response rate, 76%).

Table 1 shows participants’ demographic characteristics. The average ages were 34.52 (SD 4.71) years for patients with azoospermia and 34.95 (SD 4.00) years for patients with normozoospermia (t = −0.70, p = 0.49). The average ages of the wives in the corresponding groups were 32.76 (SD 4.32) and 33.51 (SD 4.42) years, respectively (t = −1.21, p = 0.23). There were no significant differences in the body mass index, income, educational levels, lifestyle factors (smoking status, drinking status, frequency of physical exercise) or stress levels between the two male groups (p >0.05). Educational levels (χ2 = 15.78, p = 0.00) and drinking status (χ2 = 8.11, p = 0.04) were significantly different between the two female groups (Table 1).

Table 1 Participants’ demographic characteristics.

	Male (n = 200)
Mean ± SD/n (%)				Female (n = 200)
Mean ± SD/n (%)				
Characteristics	Azoospermia
(n = 100)	Normozoospermia
(n = 100)	Statistics	Effect size
(d or V)	P1 value	Azoospermia
(n = 100)	Normozoospermia
(n = 100)	Statistics	Effect size
(d or V)	P2 value	
Age (years)	34.52 ± 4.71	34.95 ± 4.00	t =  − 0.70	d =  − 0.10	0.49	32.76 ± 4.32	33.51 ± 4.42	t =  − 1.21	d =  − 0.17	0.23	
BMI (kg/m2)	25.48 ± 3.49	25.47 ± 3.72	t = 0.03	d = 0.00	0.98	22.86 ± 3.33	23.33 ± 3.35	t =  − 1.00	d =  − 0.14	0.32	
Income(%)			χ2 = 1.02	V = 0.07	0.60			χ2 = 0.42	V = 0.05	0.81	
Low	23 (23.0)	26 (26.0)				45 (45.0)	43 (43.0)				
Middle	59 (59.0)	61 (61.0)				44 (44.0)	48 (48.0)				
High	18 (18.0)	13 (13.0)				11 (11.0)	9 (9.0)				
Education			χ2 = 3.76	V = 0.14	0.29			χ2 = 15.78	V = 0.28	0.00*	
High school and below	34 (34.0)	23 (23.0)				27 (27.0)	30 (30.0)				
College for professional training	19 (19.0)	22 (22.0)				30 (30.0)	19 (19.0)				
Undergraduate	37 (37.0)	47 (47.0)				24 (24.0)	45 (45.0)				
Postgraduate and above	10 (10.0)	8 (8.0)				19 (19.0)	6 (6.0)				
Smoker	41 (41.0)	39 (39.0)	χ2 = 0.08	V = 0.02	0.89	6 (6.0)	4 (4.0)	χ2 = 0.42	V = 0.05	0.75	
Drinking alcohol			χ2 = 3.50	V = 0.13	0.48			χ2 = 8.11	V = 0.20	0.04*	
Almost every day	2 (2.0)	1 (1.0)				0	0				
Often	7 (7.0)	4 (4.0)				2 (2.0)	1 (1.0)				
Sometimes	23 (23.0)	27 (27.0)				15 (15.0)	4 (4.0)				
Rarely	47 (47.0)	39 (39.0)				38 (38.0)	38 (38.0)				
Never	21 (21.0)	29 (29.0)				45 (45.0)	57 (57.0)				
Frequency of physical exercise			χ2 = 0.55	V = 0.05	0.76			χ2 = 1.92	V = 0.10	0.38	
None	23 (23.0)	24 (24.0)				40 (40.0)	31 (31.0)				
≤2 times a month	29 (29.0)	33 (33.0)				29 (29.0)	31 (31.0)				
≥3 times a month	48 (48.0)	43 (43.0)				31 (31.0)	38 (38.0)				
Stress levels			χ2 = 2.61	V = 0.11	0.27			χ2 = 2.27	V = 0.11	0.30	
High	43 (43.0)	32 (32.0)				30 (30.0)	27 (27.0)				
General	48 (48.0)	58 (58.0)				44 (44.0)	54 (54.0)				
Low	9 (9.0)	10 (10.0)				26 (26.0)	19 (19.0)				
Notes.

BMI body mass index

* p < 0.05.

Comparison of sexual function and behaviours between the couples with azoospermia and normozoospermia

The results of the independent t-test showed that the total IIEF-15 score (53.07 ± 11.11 vs. 57.52 ± 8.57, t = −3.17, p = 0.00) and ratings from four domains of male sexual function (erectile function [23.25 ±5.13 vs. 24.82 ± 4.18, t = −2.37, p = 0.02], orgasmic function [7.34 ± 1.82 vs. 8.04 ± 1.48, t = −2.98, p = 0.00], intercourse satisfaction [9.45 ± 2.71 vs. 10.55 ± 2.18, t = −3.16, p = 0.00], and overall satisfaction [6.96 ± 2.24 vs. 7.67 ±1.86, t = −2.44, p = 0.02]) were lower for men with azoospermia than for men with normozoospermia. In contrast, the PEDT scores (6.58 ± 3.13 vs. 5.17 ± 2.22, t = 3.67, p = 0.00) and PE incidence (χ2 = 14.73, p = 0.00) were higher for men with azoospermia than for men with normozoospermia. The chi-square test showed that the sexual satisfaction (χ2 = 7.32, p = 0.03) was lower in men with azoospermia than in men with normozoospermia. Results of the independent t-test showed that the frequency of sexual activity was lower in men with azoospermia than in men with normozoospermia (4.14 ± 2.72 vs. 5.04 ± 2.25, t = −2.55, p = 0.01). The sexual desire scores (6.07 ± 1.37 vs. 6.24 ± 1.29, t = −0.90, p = 0.37) and incidence of ED (χ2 = 6.40, p = 0.09) were not significantly different between the two groups (Table 2).

Table 2 Comparison of sexual health between men with azoospermia and normozoospermia.

Items	Azoospermic men
(n = 100)
n (%)/ Mean ± SD	Normozoospermic men
(n = 100)
n (%)/ Mean ± SD	Statistics	Effect size
(d or V)	P value	
Sexual satisfaction			χ2 = 7.32	V = 0.19	0.03*	
Satisfied	56 (56.0)	70 (70.0)				
Neutral	20 (20.0)	20 (20.0)				
Dissatisfied	24 (24.0)	10 (10.0)				
Sexual life frequency (per month)	4.14 ± 2.72	5.04 ± 2.25	t =  − 2.55	d =  − 0.36	0.01*	
IIEF-15 score	53.07 ± 11.11	57.52 ± 8.57	t =  − 3.17	d =  − 0.45	0.00**	
Erectile function score	23.25 ± 5.13	24.82 ± 4.18	t =  − 2.37	d =  − 0.34	0.02*	
Orgasmic function score	7.34 ± 1.82	8.04 ± 1.48	t =  − 2.98	d =  − 0.42	0.00**	
Sexual desire score	6.07 ± 1.37	6.24 ± 1.29	t =  − 0.90	d =  − 0.13	0.37	
Intercourse satisfaction score	9.45 ± 2.71	10.55 ± 2.18	t =  − 3.16	d =  − 0.45	0.00**	
Overall satisfaction score	6.96 ± 2.24	7.67 ± 1.86	t =  − 2.44	d =  − 0.34	0.02*	
Incidence of ED			χ2 = 6.40	V = 0.18	0.09	
No ED (26–30)	44 (44.0)	54 (54.0)				
Mild ED (22–25)	22 (22.0)	25 (25.0)				
Mild to moderate ED (17–21)	21 (21.0)	17 (17.0)				
Moderate ED (11–16)	13 (13.0)	4 (4.0)				
Severe ED (<11)	0	0				
PEDT score	6.58 ± 3.13	5.17 ± 2.22	t = 3.67	d = 0.52	0.00**	
Incidence of PE			χ2 = 14.73	V = 0.27	0.00**	
No PE (≤8)	65 (65.0)	85 (85.0)				
Probable PE (9–10)	22 (22.0)	14 (14.0)				
PE (≥11)	13 (13.0)	1 (1.0)				
Notes.

IIEF-15 International Index of Erectile Dysfunction

ED Erectile dysfunction

PEDT Premature Ejaculation Diagnostic Tool

PE Premature ejaculation

* p < 0.05.

** p < 0.01.

For the wives, the results of the independent t-test showed that the total FSFI scores (25.12 ± 5.56 vs. 26.75 ± 4.82, t = −2.22, p = 0.03) and ratings from three of the six domains of female sexual function (orgasm [4.22 ± 1.26 vs. 4.60 ± 1.03, t = −2.33, p = 0.02], satisfaction [4.10 ±1.39 vs. 4.72 ± 1.09, t = −3.51, p = 0.00], and coital pain [4.72 ± 1.06 vs. 5.04 ± 0.92, t = −2.27, p = 0.02]) were lower in the group with azoospermia than in the group with normozoospermia. The chi-square test revealed that sexual satisfaction was lower in the group with azoospermia than in the group with normozoospermia (χ2 = 14.18, p = 0.00). The results of the independent t-test indicated that the scores for sexual desire, sexual arousal ability, and vaginal lubricity were not significantly different between the two groups (p > 0.05) (Table 3).

Table 3 Comparison of sexual health between wives of men with azoospermia and normozoospermia.

Items	Azoospermia
(n = 100)
n (%)/ Mean ± SD	Normozoospermia
(n = 100)
n (%)/ Mean ± SD	Statistics	Effect size
(d or V)	P value	
Sexual satisfaction			χ2 = 14.18	V = 0.27	0.00**	
Satisfied	57 (57.0)	72 (72.0)				
Neutral	13 (13.0)	19 (19.0)				
Dissatisfied	30 (30.0)	9 (9.0)				
Sexual life frequency (per month)	3.32 ± 2.08	4.41 ± 2.91	t =  − 3.03	d =  − 0.43	0.00*	
FSFI score	25.12 ± 5.56	26.75 ± 4.82	t =  − 2.22	d =  − 0.31	0.03*	
Sexual desire score	3.39 ± 0.85	3.31 ± 0.86	t = 0.66	d = 0.09	0.49	
Sexual arousal ability score	3.84 ± 1.15	3.96 ± 1.13	t =  − 0.74	d =  − 0.11	0.46	
Vaginal lubricity score	4.85 ± 1.02	5.12 ± 0.93	t =  − 1.96	d =  − 0.28	0.05	
Orgasm score	4.22 ± 1.26	4.60 ± 1.03	t =  − 2.33	d =  − 0.33	0.02*	
Sexual satisfaction score	4.10 ± 1.39	4.72 ± 1.09	t =  − 3.51	d =  − 0.50	0.00**	
Coital pain score	4.72 ± 1.06	5.04 ± 0.92	t =  − 2.27	d =  − 0.32	0.02*	
Incidence of sexual dysfunction	33 (33.0)	21 (21.0)	χ2 = 3.07	V = 0.14	0.05	
Notes.

FSFI Female Sexual Function Index

* p < 0.05.

** p < 0.01.

Changes in sexual behaviours of the couples with azoospermia and normozoospermia during the lockdown

Regarding changes in sexual behaviours among men, the chi-square test revealed no significant differences in the perceived changes in sexual satisfaction (χ2 = 0.71, p = 0.70), sexual desire (χ2 = 0.00, p = 1.00), frequency of sexual activity (χ2 = 0.23, p = 0.89), frequency of masturbation (χ2 = 4.62, p = 0.20), or pornography use (χ2 = 2.71, p = 0.44) between the two groups (Table 4).

Table 4 Changes in sexual behaviours of couples with azoospermia and normozoospermia during the lockdown.

Items	Men				Women				
	Azoospermia
(n = 100)
n (%)/Mean ± SD	Normozoospermia (n = 100)
n (%)/Mean ± SD	Statistics	Effect size
(V)	P value	Azoospermia
(n = 100)
n (%)/Mean ± SD	Normozoospermia (n = 100)
n (%)/Mean ± SD	Statistics	Effect size
(V)	P value	
Sexual satisfaction			χ2 = 0.71	V = 0.06	0.70			χ2 = 7.22	V = 0.19	0.03*	
Increased	2 (2.0)	2 (2.0)				5 (5.0)	3 (3.0)				
Unchanged	83 (83.0)	87 (87.0)				76 (76.0)	90 (90.0)				
Decreased	15 (15.0)	11 (11.0)				19 (19.0)	7 (7.0)				
Sexual desire			χ2 = 0.00	V = 0.00	1.00			χ2 = 2.79	V = 0.12	0.25	
Increased	4 (4.0)	4 (4.0)				7 (7.0)	5 (4.0)				
Unchanged	83 (83.0)	83 (83.0)				77 (77.0)	86 (87.0)				
Decreased	13 (13.0)	13 (13.0)				16 (16.0)	9 (9.0)				
Sexual frequency			χ2 = 0.23	V = 0.03	0.89			χ2 = 1.73	V = 0.09	0.42	
Increased	4 (4.0)	5 (5.0)				7 (7.0)	4 (4.0)				
Unchanged	77 (77.0)	78 (78.0)				78 (78.0)	85 (85.0)				
Decreased	19 (19.0)	17 (17.0)				15 (15.0)	11 (11.0)				
Frequency of masturbation			χ2 = 4.62	V = 0.15	0.20			χ2 = 21.96	V = 0.33	0.00**	
Increased	12 (12.0)	6 (6.0)				24 (24.0)	2 (2.0)				
Unchanged	45 (45.0)	46 (46.0)				32 (32.0)	41 (41.0)				
Decreased	23 (23.0)	18 (18.0)				6 (6.0)	11 (11.0)				
None	20 (20.0)	30 (30.0)				38 (38.0)	46 (46.0)				
Frequency of pornography use			χ2 = 2.71	V = 0.12	0.44			χ2 = 10.90	V = 0.23	0.01*	
Increased	12 (12.0)	6 (6.0)				17 (17.0)	3 (3.0)				
Unchanged	43 (43.0)	41 (41.0)				23 (23.0)	27 (27.0)				
Decreased	7 (7.0)	8 (8.0)				4 (4.0)	5 (5.0)				
None	38 (38.0)	45 (45.0)				56 (56.0)	65 (65.0)				
Notes.

* p < 0.05.

** p < 0.01.

Among women, the chi-square test showed that the perceived changes in sexual satisfaction (χ2 = 7.22, p = 0.03), frequency of masturbation (χ2 = 21.96, p = 0.00), and pornography use (χ2 = 10.90, p = 0.01) were significantly different between the group with azoospermia and normozoospermia. However, there were no significant differences in the perceived changes in sexual desire (χ2 = 2.79, p = 0.25) or frequency of sexual activity (χ2 = 1.73, p = 0.42) (Table 4).

Psychological health of the couples with azoospermia and normozoospermia in the context of the COVID-19 pandemic

The GAD-7 scores (men: 7.18 ± 5.56 vs. 5.68 ± 4.58, t = −2.08, p = 0.04; women: 6.65 ± 5.06 vs. 5.10 ± 3.29, t = 2.57, p = 0.01) and PHQ-9 scores (men: 10.21 ± 6.37 vs. 7.49 ± 6.10, t = 3.08, p = 0.00; women: 8.81 ± 6.50 vs. 6.98 ± 4.43, t = 2.33, p = 0.02) were significantly higher for couples with azoospermia than for couples with normozoospermia. The incidence rates of anxiety, depression, and COVID-19-related anxiety were higher among couples with azoospermia than among couples with normozoospermia (p < 0.05) (Table 5).

Table 5 Psychological health of couples with azoospermia and normozoospermia in the context of the COVID-19 pandemic.

Items	Men
n (%)/Mean ± SD				Women
n (%)/Mean ± SD					
	Azoospermia
(n = 100)	Normozoospermia (n = 100)	Statistics	Effect size
(d or V)	P 1 value	Azoospermia
(n = 100)	Normozoospermia (n = 100)	Statistics	Effect size
(d or V)	P 2 value	P value	
GAD-7 score	7.18 ± 5.56	5.68 ± 4.58	t = 2.08	d =  − 0.29	0.04*	6.65 ± 5.06	5.10 ± 3.29	t = 2.57	d = 0.36	0.01*	0.01**	
Prevalence (%)	39 (39.0)	28 (28.0)	χ2 = 2.72	V = 0.12	0.10	28 (28.0)	16 (16.0)	χ2 = 4.20	V = 0.15	0.04*	0.01**	
PHQ-9 score	10.21 ± 6.37	7.49 ± 6.10	t = 3.08	d = 0.44	0.00*	8.81 ± 6.50	6.98 ± 4.43	t = 2.33	d = 0.33	0.02*	0.00**	
Prevalence (%)	43 (43.0)	30 (30.0)	χ2 = 3.65	V = 0.14	0.06	28 (28.0)	20 (20.0)	χ2 = 1.75	V = 0.09	0.19	0.02**	
COVID-19 related anxiety			χ2 = 3.78	V = 0.14	0.15			χ2 = 11.18	V = 0.24	0.00*	0.00**	
Severe	9 (9.0)	3 (3.0)				13 (13.0)	1 (1.0)					
Slight	30 (30.0)	27 (27.0)				15 (15.0)	19 (19.0)					
No	61 (61.0)	70 (70.0)				72 (72.0)	80 (80.0)					
Notes.

P1: men with azoospermia compared with normozoospermia.

P2: the wives of men with azoospermia compared with normozoospermia.

P: couples with azoospermia compared with normozoospermia.

GAD-7 Generalized Anxiety Disorder-7

PHQ-9 Patient Health Questionnaire-9

* p < 0.05.

** p < 0.01.

Association between sexual function and psychological health in couples with azoospermia

Results of the APIM showed that for couples with azoospermia, sexual function negatively correlated with their own anxiety (men: β = −0.22, standard error (SE) = 0.04, Z = −5.08, p = 0.00; women: β = −0.38, SE = 0.09, Z = −4.46, p = 0.00) and depression symptoms (men: β = −0.21, SE = 0.05, Z = −4.46, p = 0.00; women: β = −0.57, SE = 0.12, Z = −4.89, p = 0.00) but not with their partner’s anxiety (men: β = −0.00, SE = 0.04, Z = −0.07, p = 0.94; women: β = −0.07, SE = 0.09, Z = −0.77, p = 0.44) and depression symptoms (men: β = 0.01, SE = 0.05, Z = 0.25, p = 0.80; women: β = −0.12, SE = 0.11, Z = −1.08, p = 0.28). The model revealed a good overall fit (χ2/df =0.00, CFI = 1.00, RMSEA = 0.00, SRMR = 0.00, NFI = 1.00). None of the studied partner effects were significant, but the actor effects were significant (Fig. 1).

Figure 1 Association between psychological health and sexual function in couples with azoospermia using the Actor–Partner Interdependence Model.

Observed variables are shown within rectangles. Significant values are shown in red and solid line, no significant correlations in blue and dashed line. IIEF-15 represents male sexual function, and FSFI represents female sexual function. GAD-7 and PHQ-9 represent anxiety and depression symptoms, respectively. For men, IIEF-15 negatively correlated with their own GAD-7 (β = −0.22, SE = 0.04, Z = −5.08, p = 0.00) and PHQ-9 (β = −0.21, SE = 0.05, Z = −4.46, p = 0.00) values but not with their partner’s GAD-7 (β = −0.00, SE = 0.04, Z = −0.07, p = 0.94) and PHQ-9 (β = 0.01, SE = 0.05, Z = 0.25, p = 0.80) values. For women, FSFI negatively correlated with their own GAD-7 (β = −0.38, SE = 0.09, Z = −4.46, p = 0.00) and PHQ-9 (β = −0.57, SE = 0.12, Z = −4.89, p = 0.00) values but not with their partner’s GAD-7 (β = −0.07, SE = 0.09, Z = −0.77, p = 0.44) and PHQ-9 (β = −0.12, SE = 0.11, Z = −1.08, p = 0.28) values.

Discussion

This is the first study to evaluate the sexual health and psychological health of couples with azoospermia in the context of the COVID-19 pandemic. Given the rarity of the diagnosis and the delicate nature of the study, our sample of couples with azoospermia is relatively impressive. Our study found that the incidence of sexual dysfunction in couples with azoospermia was significantly higher than that in couples with normozoospermia. Couples with azoospermia experienced higher rates of anxiety and depression than couples with normozoospermia. The APIM showed that sexual function negatively correlated with the couples’ own anxiety and depression symptoms but not with their partner’s anxiety and depression symptoms.

Our study systematically compared the sexual health of couples with azoospermia as studies focusing on the sexual health of couples with azoospermia are limited (Kızılay, Şahin & Altay, 2018) and are especially lacking in data on the wives of men with azoospermia. In couples with azoospermia, the incidence of sexual dysfunction was significantly higher than that in the control group. The scores for ED, orgasmic function, intercourse satisfaction, and overall satisfaction were significantly lower for men with azoospermia than for men with normozoospermia. This is consistent with the result of a previous study, which found an association between the severity of semen quality impairment and sexual dysfunction (Lotti et al., 2016). Lotti et al.’s study (2016) found that infertile men showed more ejaculatory latency and reduced sexual health, sexual desire, and orgasmic function than fertile men. In our study, the incidence of PE in men with azoospermia was higher than that in men with normozoospermia, which is consistent with the findings of previous studies (Lotti et al., 2016; Gao et al., 2014).

Notably, there are significantly few studies on the sexual health of wives of the men with azoospermia. Our study found that the FSFI scores (orgasm, sexual satisfaction, coital pain) of the wives of men with azoospermia were significantly lower than those of the wives of men with normozoospermia. The incidence of sexual dysfunction was insignificantly higher in the wives of men with azoospermia than in the wives of men with normozoospermia (p = 0.05), which may be owing to the cut-off value (Wiegel, Meston & Rosen, 2005) and relatively small sample size. Despite this, we believe that the wives of the men with azoospermia have a lower quality of sexual function than those of men with normozoospermia, as evidenced by the FSFI scores for orgasm function and intercourse pain (p < 0.05) and a p value of 0.05 for the incidence of sexual dysfunction. Our results are consistent with those from the study by Kızılay, Şahin & Altay (2018), in which the FSFI scores of the wives of men with azoospermia were significantly lower than those of the wives of men with oligospermia or normozoospermia. Previous studies have also reported that female sexual function is significantly related to male sexual function in the general population (Jiann, Su & Tsai, 2013). In our study, the sexual function of couples with azoospermia was significantly lower than that of the control group, although the wives of men with azoospermia did not have organic diseases that affect sexual function. Female sexual function can be influenced by various factors, such as biological (hormonal and pelvic floor disorders), psychosexual (emotional and affective), and contextual (relationship discord, partner’s health problems, and sexual dysfunction) factors (Graziottin et al., 2006). Therefore, the diagnosis of azoospermia with perceived loss of masculinity and virility (Reder, Fernandez & Ohl, 2009), as well as the unfulfilled desire to conceive and psychological distress, may negatively affect women’s sexual function.

Our study is the first to investigate the changes in the sexual behaviours of couples with azoospermia during the lockdown. Our study found that there were no significant perceived changes in sexual behaviours in men with azoospermia compared with that in men with normozoospermia. However, perceived changes in sexual satisfaction, frequency of masturbation, and pornography use were more frequent among the wives of men with azoospermia than among the wives of men with normozoospermia. We observed that the rate of increased frequency of pornography use and masturbation in wives of men with azoospermia was significantly higher than that of the control group. An online survey conducted in England and Spain reported that 10% of the participants masturbated more than usual during the lockdown (Ibarra François et al., 2020). Increased masturbation frequency and frequent pornography use are related to a decline in the quality of sexual life and sexual satisfaction (Brody & Costa, 2009; Böthe et al., 2020), which is consisted with our study that the rate of decreased sexual satisfaction in the group with azoospermia was significantly higher than that of the control group. Pornography use plays a role in coping with negative moods, stress, and anxiety (Wordecha et al., 2018). Increased use of pornography is related to the need for distraction from loneliness, distress, boredom, or pandemic-related emotions (Grubbs, 2020). Therefore, the perceived changes in sexual behaviours we have observed may be related to the high incidence rate of depression and anxiety symptoms in the patients with azoospermia.

A recently conducted study reported that changes in sexual behaviours are closely related to psychological emotions (Carvalho et al., 2021). Another previous study also evaluated the moderating effects of sexual activity on mental health and sexual health during the COVID-19 pandemic (Mollaioli et al., 2021). Because our study did not utilise a pre–post repeated measures design, we do not know the mental health scores of couples with azoospermia before the COVID-19 pandemic. Large further longitudinal multicentre clinical trials should be conducted to further confirm the association between changes in sexual behaviours and mental health.

In contrast, we discovered that couples with azoospermia had higher levels of psychological distress, as evidenced by both the GAD-7 and PHQ-9 scores, than couples with normozoospermia. A recent study has reported that anxiety and depression have a significant effect on the sexual function of patients with infertility (Fernandes et al., 2020). To investigate the association between sexual function and psychological health of patients with azoospermia, we utilised the APIM. This model allows the analysis of dyads in one model (Fernandes et al., 2021), which can determine whether the sexual function of patients with azoospermia is related not only to one’s own psychological status but also to the psychological health of their partner. Our model showed that sexual function negatively correlated with patient’s own anxiety and depression symptoms but not with their partner’s anxiety and depression symptoms in couples with azoospermia. This is the first time that the APIM has been used to investigate the association between sexual function and psychological health in patients with azoospermia.

During the treatment of infertility, it is significantly important to consider not only the couple’s reproductive health but also their sexual and psychological health. It is vital to detect these issues early and treat them appropriately with sexological and psychological therapy, especially during the pandemic.

This study has some limitations. First, quantitative data about mental and sexual health prior to the COVID-19 pandemic were not collected in this study. Our data were collected only at one time point during the lockdown and therefore cannot assess changes accurately. Second, because the questionnaire was anonymous, we could not obtain the patients’ clinical data, such as information regarding azoospermia classification or serum hormone levels, which are indicators that can affect the sexual health of patients with azoospermia. However, if the questionnaires had been identifiable by name, the patients with azoospermia might have been reluctant to participate in this study or unwilling to answer truthfully. Finally, the current study included the use of self-designed questions (in reference to other studies on changes in sexual behaviours during the pandemic (Li et al., 2020a; Li et al., 2020b) and relied on self-reported cross-sectional data from local convenient samples. Thus, our findings need to be confirmed through further longitudinal multicentre clinical trials.

Conclusions

Couples with azoospermia had a lower quality of sexual function and higher levels of psychological distress than couples with normozoospermia. In couples with azoospermia, the sexual function negatively correlated with their own and depression symptoms but not with their partner’s anxiety and depression symptoms. The effects on the wives observed in the APIM are important to consider when investigating the effects on both members of the couple and not just the afflicted individual. This finding also provides a reference for patients with other causes of infertility. It is essential to provide sexual and psychological health counselling to patients with infertility, especially during the pandemic.

Supplemental Information

Supplemental Information 1 Raw data

Click here for additional data file.

Supplemental Information 2 Questionnaire for males

Click here for additional data file.

Supplemental Information 3 Questionnaire for females

Click here for additional data file.

Supplemental Information 4 Questionnaire for males (Chinese version)

Click here for additional data file.

Supplemental Information 5 Questionnaire for females (Chinese version)

Click here for additional data file.

Additional Information and Declarations

Competing Interests

Author Contributions

Human Ethics

Data Availability

The authors declare there are no competing interests.

Meng Dong and Yanqiang Tao conceived and designed the experiments, performed the experiments, analyzed the data, prepared figures and/or tables, authored or reviewed drafts of the paper, and approved the final draft.

Shanshan Wu performed the experiments, analyzed the data, prepared figures and/or tables, authored or reviewed drafts of the paper, and approved the final draft.

Zhengtao Li and Xiaobin Wang performed the experiments, authored or reviewed drafts of the paper, and approved the final draft.

Jichun Tan conceived and designed the experiments, performed the experiments, authored or reviewed drafts of the paper, and approved the final draft.

The following information was supplied relating to ethical approvals (i.e., approving body and any reference numbers):

The study protocol was approved by the Institutional Review Board for Research on Human Subjects at the Center of Reproductive Medicine (2020PS009F).

The following information was supplied regarding data availability:

The data are available in the Supplemental File.

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
