# Peer review of "Sexual and psychological health of couples with azoospermia in the context of the COVID-19 pandemic"

_PeerJ, doi:10.7717/peerj.12162_

## Round 0.1 · original submission · Major Revisions

I have been very fortunate to receive two very helpful reviews from experts in this general area. As this topic is somewhat outside my area of expertise, my inclination is to defer to their judgment. I agree with Reviewer 1 that, although there are certainly data worthy of publication here, the rationale and framing for the study needs to be made clearer. Although I can follow your logic, it feels a bit underspecified. As Reviewer 1 asks, why should these couples, in particular, be at greater risk than other couples, and especially during the pandemic compared to any other similar period of time? In other words, the fact that these data were collected during the pandemic seems almost tangential rather than central to the primary contributions of the data. Although I also agree with both reviewers about the quality of the writing, I will not spend a lot of time making line-by-line corrections at this stage as I think your paper requires significant revision. I cannot guarantee that a revision will be acceptable given the number of details that are missing from the current version. You will need to satisfactorily justify the approach you have taken to assessing covid-related ‘change.’

Reviewer 1 has helpfully uploaded an annotated PDF of your MS with some corrections and suggestions so please review that carefully.
It seems that your analyses do not take full advantage of one of the strengths of your study. You have data from both members of your couples, which should be highlighted as a unique advantage. This would allow you to examine partner effects in an APIM model rather than analyzing men and women as separate samples, which also does not allow you to examine the effects of sex on your outcomes or, more importantly, to separate individual from partner effects on sexual satisfaction. Further, your approach is somewhat hidden in the text of your MS. I would strongly encourage you to incorporate APIM analyses into your revision and to present a stronger theoretical model for your hypotheses.

From the beginning, some statements are slightly misleading.
On line 49, I don’t think you really mean to refer to the length of the disease. Many diseases like the common cold and flu have been around for a very long time. What is pertinent here is the pandemic status of the virus and the ensuing global lockdown. You need to be careful to be more precise in your language throughout the MS.
I am also not sure that the word “unprecedented” should be used twice in the first paragraph. There have of course been other pandemics. Perhaps you mean in modern times or on a global scale? But the global nature of the pandemic is also somewhat irrelevant to your research question. What is important to your study is the lockdown rather than the disease itself and the subsequent social limitations, correct? But you have not examined whether participants experienced other stressors like loss of job, isolation, illness, illness or death of loved ones, etc. The extent to which participants have been directly affected by the pandemic might be more telling than have them simply respond to a single question about perceived stress.
You need to be especially careful not to treat COVID as a causal factor in couples’ satisfaction etc. You could not manipulate and control the experience of the pandemic so any apparent consequences must be described as “associated with” not “caused by.” You will not be able to determine with certainty that any changes in responses over time were due specifically to the pandemic or to the myriad of other events that occurred simultaneously alongside it.
When you discuss previous findings, more details need to be made clear to consider the information in the appropriate context. For example, on line 70 – were these changes in frequency of intercourse in couples that lived/quarantined together or just in general? The latter would seem unsurprising due to concerns about health risks and limited opportunities for social interactions, but the former would require more explanation.

I agree with Reviewer 2’s concern about double-barreled questions as well. Although you cannot fix this now, you need to address the limitations of such an approach carefully in the discussion.
Where were the questionnaires completed? Did they complete the measures only at a single time point? It would have been beneficial of course to look at changes over time from early in the pandemic to at least 6 months or one year into the pandemic. If you asked the questions at a single time point, you are dealing with perceived change not change per se and then you need to be very specific about how you framed such questions – with what reference point? You must address the fact that couples might assume changes that didn’t actually occur because of various biases.

Participants should be described fully in a Participants section before the Results. Materials need to be described fully including subsections for each established questionnaire with alpha levels, details on scoring, number of items, sample items etc. Were these presented as pen and paper or on a computer?

I would need more information on how the two samples were equivalent, and this should appear in Participants (not results). They should at least be fairly equivalent in terms of age and length of relationship, desire to conceive, etc. Some of this information appears in Table 1 but I do not see length of relationship, which is especially relevant to sexual satisfaction and should be entered as a covariate in any analyses. What do the numbers refer to in the table for education – are those percentages rather than mean scores? For smoking status, it is not necessary to state both “yes” and “no” since they are the inverse of each other. Why did the control group men undergo a fertility examination? This suggests to me that their wives may also have experienced some difficulty conceiving, making them not a true control group (but again, I am not an expert on this topic so please feel free to correct my misunderstanding). I agree with Reviewer 1 that it needs to be clear what tests were used to compare the groups within the text where results are reported. You should indicate all p’s not just a single p here.

Your analytic approach for each hypothesis needs to be clear. As the reviewers indicate, you need to include test statistics and confidence intervals where appropriate, not just p values.

You mention mediation but it isn’t clear which variables are your mediators (line 148). You indicate later that frequency of masturbation is a mediator but it is not clear why.

It still is not clear how “changes in sexuality” were assessed (line 147). There needs to be a clear connection between hypotheses and analyses.

How are the effects of stress during the pandemic any different from any other effects of stress? In other words, how do we know that the effects here are specifically related to effects of COVID, rather than that you happened to capture a time when stress was high and stress always has a deleterious effect on psychological and sexual health? It seems that you need to account more fully for other changes such as loss of work, social activities etc. Here stressing an interaction between stress and sample would be important but if this was found, it is somewhat lost.

·

Basic reporting

• This manuscript might benefit from a more extensive report of background literature. Specifically, I felt as though the authors could have developed a stronger argument for the study’s focus on azoospermic couples, as opposed to couples who suffer from other kinds of fertility issues. Why might couples suffering from azoospermia be particularly vulnerable to the effects of the COVID-19 pandemic?

• The authors might also strengthen their rationale for focusing on azoospermic couples by discussing the prevalence of this type of fertility problem. What percentage of couples suffer from azoospermia? Are certain demographics affected more than others?

• The information presented in the introduction might also benefit from some slight reorganization. The last paragraph of the introduction discusses the impact of COVID-19 on sexual health, in general. It might make more sense to first talk about the impact of COVID-19 on sexual health, generally, and to then discuss the impact of COVID-19 on the sexual health of infertile couples, specifically.

• The authors provide a definition of azoospermia in the methods section of this manuscript, which may be helpful to readers who are unfamiliar with this condition. However, providing this definition earlier (i.e. in the introduction) might increase the readability of this manuscript.

Experimental design

• In the Participant Selection section of the paper, the authors note that “all patients [were] informed that the study… did not involve any privacy”. Is this a typo or grammatical error? If not, the authors should elaborate on this point.

• The authors state that some couples were removed from analyses due to providing inconsistent answers. The authors should explain the method by which they determined inconsistent responses.

Validity of the findings

• Under the subheading “Comparison of sexual health and sexuality between azoospermic and normozoospermic couples” the authors make several statements regarding group differences on several measures, including the IIEF-15, PEDT and FSFI. Were these the results from independent samples t-tests? If so, why are p values the only statistic reported? Means and standard deviations are reported in Table 2, but these (along with t values) should appear in-text as well.

• Similarly, the standard errors and 95% confidence intervals should be reported in-text when the authors discuss the results of their SEM.

• The authors note in the limitations section that they are unable to compare the effects of azoospermia on mental health, sexual health, and relationship satisfaction before and after the pandemic. Although it is good that the authors address this limitation, I don’t know that it is given the appropriate amount of consideration. One of the key themes of this paper is the negative effect of the pandemic on various psychological and behavioral outcomes, yet this claim was not empirically tested and the authors have no way of determining how/whether the pandemic impacted the results of this study.

• In the discussion section, the authors seem fairly confident in their interpretation of the results of their SEM; namely, that increased frequency of masturbation (which they speculate is the result of the global pandemic) leads to decreased relationship satisfaction, which in turn leads to decreased sexual health. I think it would be extremely helpful for readers if this pattern of results (assuming it was hypothesized by the authors beforehand) be laid out in the introduction section of the paper. This also relates to a broader criticism I have of this manuscript, which is that the overall narrative could be greatly clarified and strengthened in both the introduction and discussion sections of this paper. Specifically, in the introduction, the authors should clearly define each of the relevant psychological constructs, lay out the expected relationships between these constructs, and give a justification as to why they predict this pattern of relationships to emerge. Then, in the discussion, these predictions should be contrasted with the results of the statistical analyses. To summarize my point: I think that it is, at times, unclear what message the authors most wish to convey in this paper, and this message could be made clearer by revising both the introduction and discussion.

• That said, I also think the authors should take some time to consider alternate interpretations of their results. The authors did not implement an experimental design (which might also be listed a limitation). Thus, the authors should recognize that the pattern of variables laid out in their SEM is not the only possible interpretation. Sexual health was the outcome variable in the present study, but other research has suggested that a deterioration of sexual health might lead to a decrease in relationship satisfaction (e.g., Young et al., 1998), while others have proposed that, at least in men, sexual disfunction may underlie psychological distress, such as depression and anxiety (e.g., Althof, 2002).

Additional comments

I think that this manuscript presents a number of interesting findings that are absolutely worth being presented publicly. However, I have some concerns with the authors’ framing of these data, as well as some recommendations for more technical aspects of this paper in the methodology and results (noted in my comments above). I believe that these concerns should be addressed before this article is accepted for publication.

Reviewer 2 ·

Basic reporting

There are some instances throughout the article in which incorrect English is used with some grammar issues/ambiguous descriptions of study details being an issue. A thorough copyediting of the manuscript to check for such instances is recommended.

Experimental design

No comment for this area.

Validity of the findings

This is the area that requires the most improvement before publication is recommended. Most notably, details of many analyses (e.g. t-tests and ANOVAs) have been omitted from the results, and methods for scoring and coding some of the variables is questionable without a reasonable accompanying rationale. There is no mention of attempts to control for Type I error despite the use of several comparisons. These issues need to be addressed, and more detailed suggestions for revision are provided in the general comments section.

Additional comments

This article investigates an interesting and novel topic: how the COVID-19 pandemic affects azoospermic couples in terms of various relationship dynamics (e.g. sexual satisfaction, psychological distress, relationship satisfaction). Additionally, the study directly compares the relationship dynamics of azoospermic couples with normozoospermic couples. The novelty of the study and the population investigated are certainly strengths of the study. However, there are numerous methodological and statistical weaknesses that need to be addressed before publication is recommended. Most notably, details of many analyses (e.g. t-tests and ANOVAs) have been omitted from the results, and methods for scoring and coding some of the variables is questionable without a reasonable accompanying rationale. See below for more detailed suggestions for revision:

In the Introduction, line 51, it should read as “It has had an unprecedented impact ON individual psychological and relationship health.”

In the Participants section, in line 92, it should read as “Before participating, all patients WERE informed that the study was voluntary….”. Also, in that same sentence, the phrasing “did not involve any privacy” does not make sense. Are the authors trying to say that it did not involve any threats to privacy? This requires clarification.

In the Self-reported Questionnaires section, in lines 111 – 112, the item the authors set up: “How satisfied are you with intimacy and sexual activity with your wife (husband)?” is double-barreled, which makes it more difficult to interpret the meaning of the responses. After all, intimacy is a construct that is distinct from sexual activity. This should at least be mentioned as a limitation. Similarly, in lines 133 - 134, the authors created another double-barreled item which would be associated with the interpretability issues: “Do you have difficulty with or pain during sexual intercourse (dyspareunia)?” Once again, “difficulty” might not always involve pain, so the exact construct being measured here is ambiguous.

In the Statistical Analysis section, in lines 144 – 145, be sure to mention that the t-test is an independent-samples t-test.

In the Results section, it would be easier for the reader to follow along if the authors state which analysis was done for each of the comparisons being made (t-test, chi-square, or ANOVA).

There are many comparisons being referenced in the Results section and no mention of attempts to control for Type I error. Were any procedures put in place to reduce Type I error? If so, please describe those. If not, explain the rationale for not implementing them.

Also, in the Results section, some type of effect size measure for the various comparison analyses (such as Cohen’s d and/or partial eta squared) would also help with interpretation of the results, especially in the case that Type I error was not controlled for.

What was the rationale for the high rate of categorization of variables that often are treated as continuous? For instance, the QMI is typically measured in a continuous fashion, with higher scores indicating higher satisfaction. However, the authors categorize couples into “fine, general, or deteriorate.” How are these categorizations determined? Furthermore, why use such categorization? Typically, creating arbitrary cutoff points in continuous scales is associated with decreased quality of analyses.

In the SEM analyses, the coding for frequency of masturbation is not intuitive, with increased masturbation coded as 0 and 3 coded as none. This makes it confusing for the reader to interpret the figures depicting these analyses with frequency of masturbation negatively predicting psychological distress and positively predicting relationship satisfaction. Consider recoding for improved interpretability.

In the Discussion, in line 334 “healthing” is not a word. Please revise.

In the Discussion, line 347, “…compare the situation…” should be changed to “…comparing the situation…”

In the Discussion, in lines 350 – 351. The privacy concern is reasonable when speaking of reasons for not including measures such as serum hormone levels and azoospermia classification. However, couldn’t there be some type of code number assigned to participants’ clinical data to retain privacy?

---

## Round 0.2 · Major Revisions

Both reviewers that previously reviewed your manuscript reviewed your revision and deemed it acceptable. However, I still have concerns with your framing and statistical approach.

I think you need to be more careful when discussing ‘changes” due to the lockdown (e.g., line 95) because you didn’t measure changes. Because you are comparing two groups (azoospermatic and normozoospermatic couples) at a single time-point, the study should be framed as a non-equivalent groups quasi-experimental design rather than as a pre-post repeated measures design, which you suggest with such misleading statements. The entire connection to COVID is really loose and tangential and should not be stressed so heavily. You happened to assess these things at a time point during COVID but you might have obtained the same results at any other time point as well.

Now that you have responded to the questions from the first round of reviews and I have a better understanding of your aims and design, I have further concerns about your analytic strategy. It seems to me that you hypothesized that azoospermatic couples would experience more stress, anxiety etc. as a result of the lockdown compared to normozoospermatic couples, which in turn would lead to greater dissatisfaction in the relationship. So you should have tested a mediational model with psychological distress as the mediator between group (couple designation) and couple satisfaction. It needs to be more clear how the constructs are operationalized and which measure represents which variable. Again, you have to be careful to discuss current stress at a time-point in the middle of the pandemic rather than a change in stress, which you did not measure. At best, you can describe perceived changes in stress because of the way the questions were framed.

There are still some places where you need to remove causal language (e.g., line 113, “had an impact on.” You cannot refer to the diagnosis of azoospermia having an impact on something because there could be other correlates of the condition that cause the outcome you’re observing. You cannot control assignment to this condition in a way that would allow you to rule out confounds.
When you make comparison statements, you must be absolutely explicit about the point of comparison. For example, on line 115, “led to increased changes” relative to what? In comparison to the control group?

You need to be more transparent about the control group. If all participants were recruited from a fertility clinic, then likely all couples were experiencing fertility concerns – not just the azoospermatic group. So, why would we expect the groups to differ in their satisfaction and response to the lockdown? Is it the fact that they have evidence that their fertility issue won’t resolve? This needs to be addressed.

You did not follow my previous advice in describing your measures with separate subheadings, sample items, reliability, and information about scoring. It needs to be very clear which measures represent predictors and outcomes and which constructs they’re being used to assess. I am still surprised when I read that you focus on masturbation frequency as the sole measure of sexual behavior.
You could highlight more that, given the rarity of the diagnosis and the delicate nature of the study, your sample of azoospermatic couples is quite impressive.

Your justification for including masturbation frequency in your path analyses seems post-hoc. You shouldn’t run preliminary analyses and then just include whatever variable significantly differs between the groups in your primary analysis. Your analyses should be hypothesis driven a-priori. In reading your paper again, I assume that you are examining various measures of sexual health as a mediator to predict psychological well-being such that the group is your main IV, their sexual behavior scores are mediators and their psychological well-being scores are outcomes. Earlier, I felt that psychological distress was the mediator. But it is still difficult to determine your exact theoretical model and how it is tested statistically. I don’t understand your conceptual models. Figures 3 and 4 are pretty pentagons but I’m not sure they make sense.

I still strongly believe that you should use an APIM model to examine partner effects. The frequency of masturbation likely impacts the partner’s sexual satisfaction as well especially if masturbation replaces sexual intimacy. As I noted previously, you have done the difficult part in gathering all of the data from both members of your couples. It would be a terrible shame not to explore these data to their fullest by examining both actor and partner effects, which the APIM model allows you to do.

There are too many tables. Revamping your statistical approach should help reduce the number of data points to tell a more concise and coherent story.

On lines 328-330, I would not word this as if the diagnosis has an impact on wives’ sexual health. It is more nuanced than this – being paired with a partner with whom one cannot have a child may impact women’s health whether they receive a formal diagnosis or not. There may be other aspects of their partners that influence their own satisfaction.

The paragraph beginning on line 371 should be integrated more with the rest of the discussion.

On line 374, the phrase “increased changes in sexual life..” doesn’t really make sense. The phrasing on lines 381-382 is still unclear. You need to explicitly admit that you have data only at one time point during and therefore cannot assess changes accurately.

·

Basic reporting

No comment for this area.

Experimental design

No comment for this area.

Validity of the findings

No comment for this area.

Additional comments

No comment for this area.

Reviewer 2 ·

Basic reporting

The authors have addressed the the reviewers' comments and concerns regarding the manuscript.

Experimental design

No comment here.

Validity of the findings

No comment here.

---

## Round 0.3 · Minor Revisions

Thank you for being receptive to the previous comments. I think the paper is much improved with clearer goals and analyses. The ability to examine both actor and partner effects is a strong novel contribution to the literature. I will be happy to accept the manuscript when you have addressed the following very minor things. The effects you observe for the wives should make it clear that it is important to investigate effects on both members of the couple and not just the afflicted member. This could be emphasized in your conclusion.

Any comments refer to the line numbers of the clean reviewing PDF.
Could you add the word “perceived” before “changes in sexual satisfaction” on line 41 and elsewhere to make it clear that you asked once about perceptions of change rather than calculating change between scores reported at two time points.

Line 79, move the “only” to after “obtained.”

In Figure 1, can you clearly indicate the actor versus partner effects in the model?

---

## Round 0.4 · accepted · Accept

Thank you very much for your attention to these final minor details.